## Reviews and syntheses: Heterotrophic fixation of inorganic carbon – significant but invisible flux in environmental carbon cycling

Alexander Braun[1], Marina Spona-Friedl[1], Maria Avramov[1], Martin Elsner[1,2], Federico Baltar[3], Thomas Reinthaler[3], Gerhard J. Herndl[3,4] & Christian Griebler[1,3]*

[1] Helmholtz Zentrum München, Institute of Groundwater Ecology, Ingolstaedter Landstrasse 1, D-85764 Neuherberg, Germany

[2] Technical University of Munich, Department of Analytical Chemistry and Water Chemistry, Munich, Germany

[3] University of Vienna, Department of Functional and Evolutionary Ecology, Althanstrasse 14, 1090 Vienna, Austria

[4] Department of Marine Microbiology and Biogeochemistry, Royal Netherlands Institute for Sea Research, Utrecht University, PO Box 59, 1790 AB Den Burg, The Netherlands

* Author for correspondence: christian.griebler@univie.ac.at

### Abstract

Heterotrophic $CO_2$ fixation is a significant, yet underappreciated $CO_2$ flux in environmental carbon cycling. In contrast to photosynthesis and chemolithoautotrophy – the main recognized autotrophic $CO_2$ fixation pathways - the importance of heterotrophic $CO_2$ fixation remains enigmatic. All heterotrophs – from microorganisms to humans – take up $CO_2$ and incorporate it into their biomass. Depending on the availability and quality of growth substrates, and drivers such as the $CO_2$ partial pressure, heterotrophic $CO_2$ fixation contributes at least 1-5% and in the case of methanotrophs up to 50% of the carbon biomass. Assuming a standing stock of global heterotrophic biomass of 47-85 Pg C, we roughly estimate that up to 5 Pg C might be derived from heterotrophic $CO_2$ fixation and up to 12 Pg C $yr^{-1}$ originating from heterotrophic $CO_2$ fixation are funneled into the global annual heterotrophic production of 34-245 Pg C $yr^{-1}$. These first estimates on the importance of heterotrophic fixation of inorganic carbon indicate that this pathway should be incorporated in present and future carbon cycling budgets.

**Key words:** $CO_2$ fixation, heterotrophs, anaplerosis, carbon cycling

## 1. Introduction

Fixation of $CO_2$ is a fundamental biosynthetic process in nature (Beer et al. 2010, Berg et al. 2007) providing the main source of metabolic energy on Earth (Giovannoni and Stingl 2005). At the same time, it acts as a sink for atmospheric $CO_2$, the most important greenhouse gas, which is responsible for more than 60% of the 'enhanced greenhouse effect' resulting in global warming (Beer et al. 2010, Berg 2011, Houghton 2007, Le Quéré et al. 2016).

While photosynthesis and chemosynthesis are the most important processes of carbon fixation, non-autotrophic carbon fixation, i.e., the carbon fixation mediated by heterotrophic organisms might also be relevant albeit uncommonly quantified. While heterotrophs are, per definition, organisms that respire organic compounds to gain energy and build up biomass, $CO_2$ fixation plays also an essential role in heterotrophic carbon metabolism. The diversity of carboxylating enzymes in nature reaches far beyond autotrophy and virtually all heterotrophs harbor numerous enzymes fixing dissolved inorganic carbon. Even though the first carboxylase in heterotrophs was discovered already more than 80 years ago (Wood and Werkman 1936), the role of heterotrophs in carbon cycling has so far largely focused on the oxidation of organic substrates using oxygen or alternative electron acceptors (e.g. nitrate, ferric iron, sulfate) and the production of $CO_2$. Similar to the $CO_2$ fixation by autotrophs, "heterotrophic $CO_2$ fixation" might, however, constitute a significant carbon flux in specific habitats. The relevance of this process has hardly been quantified due to the lack of reliable estimates of heterotrophic $CO_2$ fixation for most organisms and habitats, and the presumption that $CO_2$ fixation in natural environments is restricted to autotrophic organisms.

To fill this gap, we review the current knowledge on (i) the role of heterotrophic $CO_2$ fixation for cellular metabolism, (ii) respiration and non-autotropic $CO_2$ fixation, (iii) $CO_2$ fixation in habitats dominated by heterotrophs, and provide (iv) quantitative estimates of heterotrophic $CO_2$ fixation in different environments.

## 2. Role of heterotrophic $CO_2$ fixation for cellular metabolism

The non-autotrophic uptake of inorganic carbon has been reported for a wide range of organisms from prokaryotes and fungi to vertebrates (Woods & Werkman 1938, Kleiber et al. 1952, Cochrane 1958, Hartman et al. 1972, Perez & Matin 1982, Schinner et al. 1982, Parkinson et al. 1990, Roslev et al. 2004, Hesselsoe et al. 2005, Feisthauer et al. 2008, Spona-Friedl et al. 2020) and plants (Melzer and O'leary 1987). Currently, more than twenty carboxylases are known forming an integral part of the central and peripheral metabolic pathways of heterotrophic metabolism (Fig. 1), e.g., in gluconeogenesis, the synthesis of fatty acids, amino acids, vitamins and nucleotides, the assimilation of leucine, and in anaplerosis (Evans and Slotin 1940, Krebs 1941, Wood and Werkman 1941, Werkman and

Wood 1942, Kornberg and Krebs 1957, Wood and Stjernholm 1962, Kornberg 1965, Scrutton 1971, Hartman et al. 1973, Dijkhuizen and Harder 1985, Parkinson et al. 1991, Attwood 1995, Han et al 2000, Sauer and Eikmanns 2005, Erb et al. 2009, Schink 2009, Erb 2011, Bar-Even et al. 2012). Carboxylation in heterotrophs not just compensates for the dependence on organic matter, rather $CO_2$ fulfills the role of a "co-substrate" providing an effective and simple way to extend an existing organic carbon substrate by a single C1 unit as part of the secondary production (Erb 2011).

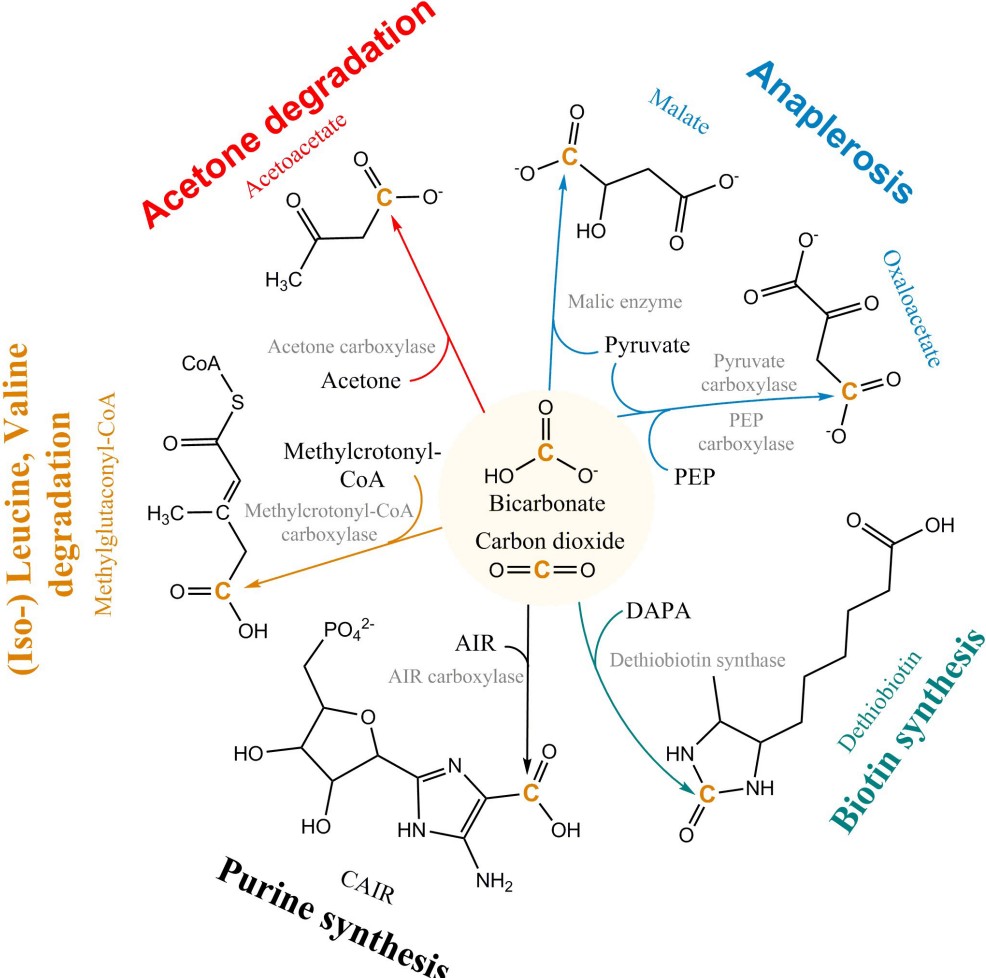

**Figure 1:** Selected heterotrophic $CO_2$ fixation reactions and pathways. PEP: phosphoenolpyruvate, DAPA: 7,8-diaminononanoate, AIR: 1-(5'-phosphoribosyl)-5-aminoimidazole, CAIR: 1-(5-phospho-D-ribosyl)-5-amino-4-imidazolecarboxylate, CoA: Coenzyme-A.

The most important $CO_2$ fixation pathway in all organisms is anaplerosis. Anaplerosis replenishes intermediates in the tricarboxylic acid (TCA) cycle, which have been released for biosynthesis. TCA metabolites are used as building blocks for macromolecular compounds, e.g. almost half of all amino acids in prokaryotes are directly synthesized from oxaloacetate and α-ketoglutarate (Fuchs 1999). For this purpose, heterotrophs use the enzymes pyruvate carboxylase present in a large variety of organisms, including prokaryotes, archaea, yeasts,

fungi and higher organisms (e.g. mammals), and phosphoenol pyruvate (PEP) carboxylase, widely distributed in bacteria (Attwood 1995; Jitrapakdee and Wallace 1999; Sauer and Eikmanns 2005; Jitrapakdee et al. 2008) (Fig. 1). The replenishment of metabolites continuously withdrawn from the TCA cycle via the anaplerotic reaction of PEP carboxylase entails an assimilation of $CO_2$ corresponding to 25% of the initial substrate's carbon content. In a systematic stable isotope labelling experiments with *Bacillus subtilis*, a gram-positive heterotrophic bacterium widespread in the environment, the interdependency of pathways and rates of $CO_2$-fixation on the concurrent utilization of organic substrate(s) was explored (Spona-Friedl et al. 2020). Over the course of the experiments *B. subtilis* assimilated 6% and 5% of carbon biomass from the external $H^{13}CO_3$ pool when growing on glucose and lactate, respectively (Spona-Friedl et al. 2020). Growth on malate, an intermediate of the TCA cycle, expected to serve directly to refill the oxaloacetate pool of the TCA cycle, still revealed a contribution to biomass production from inorganic carbon of 3% (Spona-Friedl et al. 2020). PEP carboxylase was still actively transforming pyruvate to oxaloacetate. Heterotrophic $CO_2$-fixation continued to a lower extent even in the absence of cell growth during the stationary phase (Spona-Friedl et al. 2020), indicating that anaplerotic reactions are important in low-productivity habitats (see below).

Overall, heterotrophic $CO_2$ fixation via anaplerosis in microorganisms contributes around 1 to 8% to the carbon biomass (Romanenko 1964, Perez and Matin 1982, Doronina and Trotsenko 1984, Miltner et al. 2004, Roslev et al. 2004, Hesselsoe et al. 2005, Sandruckova et al. 2005, Feisthauer et al. 2008, Akyniede et al. 2020, Spona-Friedl et al. 2020). Under particular environmental conditions even higher contributions were reported (Perez and Martin 1982). The advantage that $CO_2$ is readily available to the cell either as atmospheric gas or, more commonly, in its hydrated form $HCO_3^-$, obviously outcompetes the disadvantage that carboxylation is generally an endergonic reaction (Faber et al. 2015). This thermodynamic obstacle may be less important when carboxylation supports the assimilation of organic substrates more reduced than the organism's biomass, resulting in carbon-limited but excess-energy conditions (Heijnen and Roels, 1981, Ensign et al. 1998, von Stockar et al. 2006, Battley 2013). In this case, in addition to anaplerosis further carboxylation reactions are induced (Fig. 1) to add oxidized C (from $CO_2$) to the reduced organic substrate for adjusting the degree of reduction to that of the biomass (Fig. 2). For example, the assimilation of leucine and propionate into biomass entails carboxylation of the initial C-6 and C-3 carbon bodies, respectively and thus, triggers an assimilation of dissolved inorganic carbon (DIC) that corresponds to 17% and 33% of the initial substrate's carbon content, respectively (Erb 2011). In aerobic methane oxidation, the full oxidation potential of one molecule of $CO_2$ is needed to adjust the high degree of reduction of methane to that of biomass during its assimilation. Consequently, methanotrophs derive up to 50% of their carbon biomass from $CO_2$ (Strong, et al. 2015, Battley 2013).

126

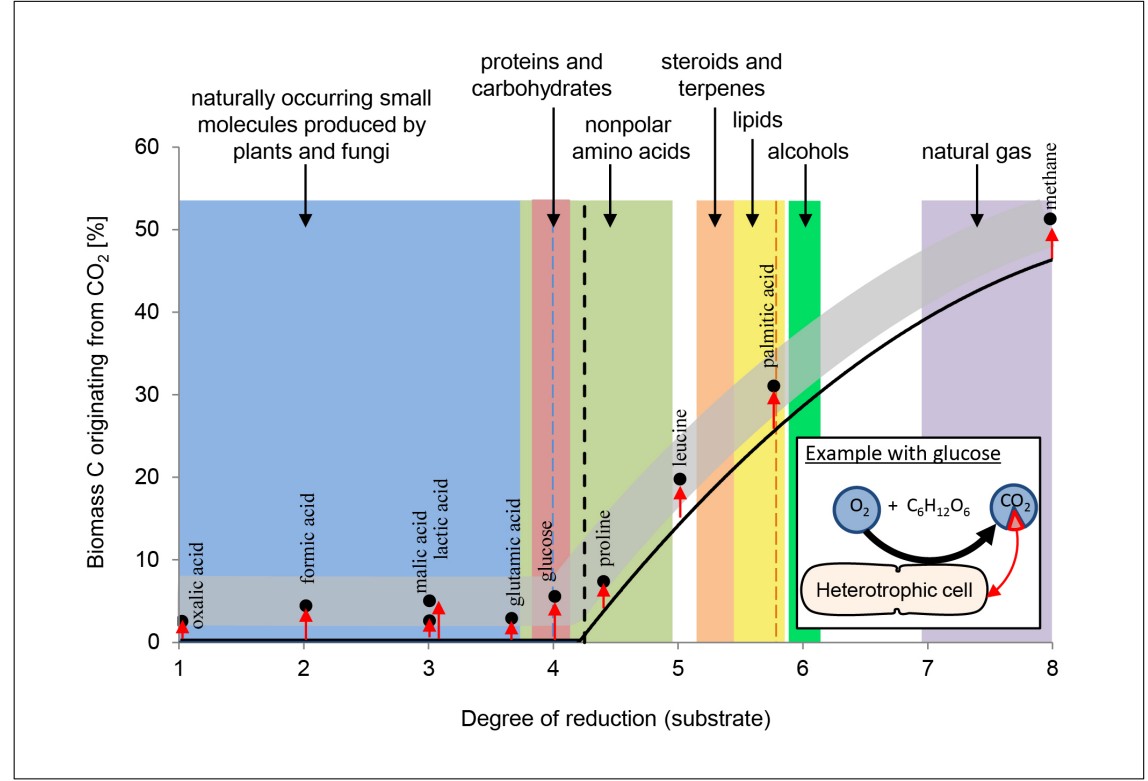

127

**Figure 2:** Anaplerotic $CO_2$ fixation contributes 1-8% of carbon biomass (indicated by the grey band) in heterotrophic cells. Dependent to the organism and in relation to the uptake of the individual organic compounds and their entry into the TCA cycle and central metabolic pathways the relative amount of inorganic carbon assimilated varies, as highlighted by the red arrows. See examples for malic and lactic acid. With organic carbon sources more reduced than the organism's biomass (dashed line) further carboxylation reactions are induced (indicated by black line), increasing the overall carbon contribution from $CO_2$ beyond anaplerosis (grey band). In methanotrophs, 50% of the cell's carbon may originate from $CO_2$ fixation. For further explanations, see text.

136

Besides the degree of reduction of organic carbon sources, the partial pressure of $CO_2$ plays a role. Carboxylases may catalyze carboxylation as well as decarboxylation of organic compounds and the equilibrium of the reaction depends on the concentrations of all compounds involved. An increase in the $CO_2$ concentration may move the equilibrium of the reaction toward the product of the carboxylation, and thus leading to an increase in $CO_2$ fixation.

In a physiological context, the amount of inorganic carbon fixed by heterotrophs, either from an endogenous or exogenous source, may be dependent on the metabolic state of the organisms and the specific environmental conditions. In their early work, Romanenko et al. (1972) suggested that the rate of heterotrophic anaplerotic fixation of DIC is strictly proportional to the heterotrophic bacterial carbon production. Since then, a number of

factors have been identified potentially influencing the relative contribution of anaplerotic and other non-autotrophic $CO_2$ fixation reactions on biomass production. In laboratory experiments with the bacterial strain *Thiobacillus novellus*, for example, a higher amount of $CO_2$ was fixed under nutrient limited conditions (Perez and Matin 1982). Moreover, mixotrophic bacterial strains fixed more DIC compared to those grown autotrophically (Perez and Matin 1982). Fungi fixed relatively more $CO_2$ at lower organic carbon (glucose and maltose) concentrations (Schinner et al. 1982). The degree of heterotrophic $CO_2$ fixation highly depended on the availability of easy degradable organic carbon sources (Schinner et al. 1982).

Studies on the possible relationship between heterotrophic DIC fixation and the activity of prokaryotic cells revealed contradicting results. While Roslev et al. (2004) mentioned actively growing cells fix more DIC than resting cells, Merlin et al. (2003) report enhanced uptake of DIC by heterotrophic bacteria during slow growth and starvation. A relationship between DIC and heterotrophic bacterial production has been reported frequently as exemplified below.

## 2. Respiration and non-autotropic $CO_2$ fixation

The production of $CO_2$ via respiration and the parallel fixation of $CO_2$ in heterotrophs take place simultaneously. The heterotrophic fixation of $CO_2$ is thus generally considered a back-reaction, i.e., part of the originally produced $CO_2$ from respiration is re-assimilated. Following this line of arguments, the more reduced an organic substrate is the less $CO_2$ is released (Fig. 2). Heterotrophic fixation of DIC does not necessarily lead to a net carbon biomass production, however, if microbes oxidize geogenic methane, this would result in a net carbon biomass production. Experimentally it is difficult to differentiate respiratory $CO_2$ flux from concurrent anaplerotic $CO_2$ fixation. As a consequence, there are numerous experiments and field studies determining dark $CO_2$ fixation, but only a few studies quantified the assimilation of DIC by non-autotrophs.

Respiration in aquatic systems is frequently determined via the consumption of dissolved oxygen (Robinson and Williams 2005) potentially underestimating the carbon use efficiency of heterotrophs. Depending on the substrate, the respiration quotient ($\Delta CO_2/-\Delta O_2$) varies between $0.7 - 1.3$ (Robinson 2019) leading to an error between 20 and 40% with regard to $CO_2$ production from respiration. Moreover, the respiration quotient also varies because other oxygen consuming processes are potentially taking place simultaneously (e.g. nitrification) (Robinson 2019). For instance, it is 138 $O_2$ for 106 $CO_2$ for ideal Redfield type organic matter, and 150 $O_2$ for 106 $CO_2$ for more realistic marine organic matter (Fraga et al. 1998; Paulmier et al. 2009). Calculations based on a study on temperate forest soils revealed a reduction of overall $CO_2$ emissions due to dark $CO_2$ fixation by mainly

heterotrophic microbes (Akinyede et al. 2020). Collectively, with respect to C cycling, heterotrophic $CO_2$ fixation and the carbon flux from the inorganic pool into heterotrophic biomass can be regarded as a process more important than hitherto assumed.

## 3. $CO_2$ fixation in habitats dominated by heterotrophs

In contrast to sunlit habitats, where photoautotrophs make up a significant portion of the total biomass and photosynthesis is of major importance in carbon cycling, heterotrophs and chemolithoautotrophs represent the prevailing biota in the "dark habitats", i.e., soils, subsurface environments and the deep sea. These dark environments, although characterized by disproportionally lower biological activity, exceed their photic counterparts in both, volume and biomass. In the oceans, the deep sea (below 200 m) exceeds the sunlit surface layer by a factor of 18 in volume and, remarkably, by a factor of two in biomass (Arístegui et al. 2009). Therefore, the so-called "dark $CO_2$ fixation" does not only occur in specific 'hot spots' on the seafloor (hydrothermal vents, cold seeps and mud volcanoes), or in anoxic waters, but also throughout the entire oxygenated 'dark' water column (Reinthaler et al., 2010, Yakimov et al., 2014). In limnic environments, the dark groundwater ecosystems outnumber surface waters 100-fold in terms of water volume (Danielopol et al. 2003), and similarly, also soils are with the exception of their surface exclusively dark habitats.

Yet, heterotrophic $CO_2$ fixation does not occur only in the dark environments since heterotrophs are also found in the photic zone. This is particularly relevant in the ocean because the photic zone is where the highest biomass concentrations are found. Recently, it has been estimated that the inclusion of dark $CO_2$ fixation (integrated over the euphotic layer, 0-150 m depth) would increase oceanic primary production estimates by 2.5–22 % (Baltar et al., 2019). A similar situation might be assumed for surface inland waters, however, global estimations are missing so far.

Dark DIC fixation has been reported for all types of ecosystems, including marine habitats (Wuchter et al. 2003, Middelburg 2011, DeLorenzo et al. 2012, Molari et al. 2013, Baltar and Herndl 2019, Lengger et al. 2019, Smith et al. 2019, Vasquez-Cardenas et al. 2020), brackish and freshwater systems (Bräuer et al. 2013, Santoro et al. 2013, Noguerola et al.2015, Signori et al. 2017, Vick-Majors and Priscu 2019, Zhao et al. 2020), cave waters and groundwater ecosystems (Pedersen & Ekendahl 1992a, 1992b; Kotelnikova & Pedersen 1998, Kellermann et al. 2012, Lazar et al. 2017), and soil habitats (Ehleringer et al. 2000, Miltner et al. 2004, 2005, Šantrůčková et al. 2005, 2018, Akinyede et al. 2020 and references therein). In the absence of solar radiation, particularly in the dark ocean, $CO_2$ fixation rates of up to ~125mg C m$^{-3}$ d$^{-1}$ have been measured, amounting to 30% (on a per volume basis) of the phototrophic $CO_2$ fixation in ocean surface waters (Zopfi et al. 2001, Detmer et al. 1993, Casamayor et al. 2001, Baltar et al. 2010). In a eutrophic lagoon, dark DIC fixation

accounted for 31% of total DIC fixation in the water column (Lliros et al. 2011). Recently it was shown that the ratio between dark/light $CO_2$ fixation in oceanic surface waters which is usually around 0.1 increases with depth reaching a ratio of 1 at 120-160 m depth (Baltar et al., 2019). In the past, however, dark DIC fixation has frequently been attributed to the activity of chemoautotrophs only. Only a few studies so far provided strong quantitative evidence for heterotrophic $CO_2$ fixation in aquatic and terrestrial ecosystems (Tab. 1).

As indicated, part of the dark $CO_2$ fixation in oceans has been attributed to chemolithoautotrophic archaea (Wuchter et al. 2003, Ingalls et al. 2006) obtaining the energy required for the endergonic carboxylation through the oxidation of reduced inorganic compounds, such as ammonia or hydrogen sulfide (Swan et al. 2011; Zhang et al. 2020). A total annual chemolithoautotrophic $CO_2$ fixation rate of 0.77Pg C was calculated for the oceans (Middelburg 2011). The observed fluxes of the reduced inorganic compounds available as energy sources, however, seem largely insufficient to explain the relatively high dark $CO_2$ fixation rates (Overbeck 1979, Tuttle and Jannasch 1979, Baltar et al. 2010, Reinthaler et al. 2010, Herndl and Reinthaler 2013). In some cases, the supply rates of the reduced inorganic compounds used as an energy source explain less than 40% of the observed dark $CO_2$ fixation rates (Zopfi et al. 2001). Recently, chemoautotrophic nitrification was estimated to explain <13% of the dark $CO_2$ fixation (integrated over the euphotic zone) with the rest coming from either heterotrophic DIC fixation or other chemoautotrophic processes (Baltar and Herndl 2019).

The potential energy sources for the unexplained proportion of the dark $CO_2$ fixation remain enigmatic. Possible explanations could be either an underestimation of the supply rates of reduced inorganic compounds or the uptake of $CO_2$ by heterotrophic organisms (Zopfi et al. 2001, Baltar et al. 2019). In the surface ocean in particular, DIC incorporation via anaplerotic reactions might play an important role in compensating metabolic imbalances in marine bacteria under oligotrophic conditions, contributing > 30 % of the carbon incorporated into biomass (González et al. 2008; Palovaara et al., 2014). Evidence for the latter comes from experiments with Arctic seawater, which exhibited high DIC fixation rates (0.5–2.5 µg C $L^{-1}$ $d^{-1}$) correlating with heterotrophic bacterial production (Alonso-Sáez et al. 2010). Using different molecular tools, DIC uptake was attributed mainly to heterotrophic *Gamma-* and *Betaproteobacteria* rather than to typical chemoautotrophs, thus showing that chemolithoauthotrophs were not the main drivers of $CO_2$ fixation in this habitat (Alonso-Sáez et al. 2010). Further evidence comes from the genome of *Polaribacter* sp. MED152, a representative of Bacteroidetes, which typically comprise about 10–20% of the prokaryotic abundance in seawater (González et al. 2008). A unique combination of membrane transporters and carboxylases in these organisms indicates the importance of anaplerosis besides other DIC fixation pathways (González et al. 2008). If the heterotrophic metabolism of bacteria is suddenly intensified (e.g., after an input of organic matter), dark DIC fixation rates and the expression of transcripts associated with key anaplerotic enzymes increase

proportionally (Baltar et al., 2016). As mentioned above, contradicting results were obtained on the relationship between heterotrophic $CO_2$ fixation and the availability of organic matter. A few studies suggest a relative increase in dark DIC fixation in oligotrophic habitats harboring slow-growing or starving bacterial populations (Perez and Matin 1982, Schinner et al. 1982, Merlin et al. 2003, Alonso-Sáez et al. 2010, Santoro et al. 2013). Considering the slow community-wide specific growth rates of heterotrophic bacteria in oligotrophic and/or cold waters, such as the marine aphotic zone, the Arctic Ocean, deep sea sediments, groundwater systems and the terrestrial subsurface, alpine limnic systems and deep-lake sediments, enhanced anaplerotic DIC uptake can be expected. However, there is also evidence for the stimulation of dark DIC fixation in response to organic matter enrichment in different types of soils (Miltner et al. 2005, Šantrůčková et al. 2018). Hence, these contradictory findings require further, more systematic research.

Other environmental factors that may influence dark DIC fixation include the concentrations of $CO_2$ and bicarbonate as inorganic carbon sources. An increase in the $CO_2$ concentration may shift the equilibrium of the carboxylation-decarboxylation reactions increasing $CO_2$ fixation. Elevated partial pressure of $CO_2$ might stimulate dark DIC fixation. In temperate forest soils, rates of dark microbial $CO_2$ fixation were positively correlated with the $CO_2$ concentration (Spohn et al. 2019). Similarly, with increasing $CO_2$ concentrations, higher dark DIC fixation was observed in wetland soils affected by subcrustal $CO_2$ degassing (Beuling et al. 2015). Here, besides known chemoautotrophs, $CO_2$ fixation via anaplerotic reactions was shown for putatively heterotrophs, i.e., subdivision 1 Acidobacteriaceae, lacking enzymatic pathways for autotrophic $CO_2$ fixation (Beuling et al. 2015). In experiments with two marine heterotrophic bacterial isolates, elevation of $CO_2$ concentration provoked an increase in $CO_2$ fixation along with a decrease in respiration (Teiro et al. 2012). Thus, we may assume that a rise in $CO_2$ concentrations and $CO_2$-induced geochemical changes will alter carbon turnover in affected ecosystems with dark DIC fixation and anaplerotic reactions becoming more important.

**4. Quantitative estimates of heterotrophic $CO_2$ fixation in different environments**

*Quantification of heterotrophic DIC fixation*

It is difficult to properly quantify heterotrophic fixation of inorganic carbon in the environment. Not surprisingly, quantitative data almost exclusively originate from laboratory experiments using cultures and tissues in combination with carbon isotopic labeling (e.g. Spona-Friedl et al. 2020). Field studies generally report on dark carbon fixation, including the activity of chemoautotrophs and heterotrophs. So far, evidence for the significant contribution of heterotrophic $CO_2$ fixation, as highlighted for selected studies in Tab. 1, is based on additional measures complementing the quantification of dark carbon


fixation, i.e. molecular microbial community analysis occasionally including the quantitative
assessment of functional genes involved in carbon fixation and characterization of
environmental conditions. In fact, significant dark carbon fixation (i) in the obvious absence
of chemoautotrophs and related genes or (ii) in well oxygenated environments that lack
potential electron donors (e.g. $H_2$, $H_2S$, $NH_4$), led authors to conclude on the quantitative
importance of heterotrophic carbon fixation (e.g. Miltner et al. 2005; Alonso-Saéz et al.
2010; Yakimov et al. 2014; Šantrůčková et al. 2018; Akinyede et al. 2020). Individual studies
succeeded to follow the label of inorganic carbon in environmental samples into amino
acids of microorganisms (Spohn et al. 2019; Spona-Friedl et al. 2020). In future, the
combined application of metabolomics and isotope tracing may help further developing this
field of research.

*Heterotrophic $CO_2$ fixation in different habitats*
Measurements of dark DIC fixation with a strong evidence of a significant contribution of
heterotrophic assimilation of DIC are scarce. In Table 1, we provide a compilation of studies
conducted in soils, marine and limnic ecosystems. Where possible, we compared dark DIC
fixation rates with heterotrophic activity. In marine and limnic systems, heterotrophic
carbon production as a widely applied activity measurement was used. In soils, we
compared dark DIC fixation rates with respiration, i.e., $CO_2$ production. Dark DIC fixation
rates in different marine systems range between 0.1 and 206 µg C $L^{-1}$ $d^{-1}$ with highest values
found in a eutrophic lagoon and lowest values in the deep waters of the Mediterranean Sea
(Tab. 1). Data from limnic systems originate from lake sediments with dark DIC fixation rates
between 0.12 and 48 mg C $m^{-2}$ $d^{-1}$ (Tab. 1). Projecting these numbers to only the top 10 cm
of sediment in the different lakes (which is a gross simplification), values of 1.2-480 µg C $L^{-1}$
sediment $d^{-1}$ are obtained. When compared to rates of bacterial carbon production, dark DIC
fixation rates in these habitats accounted for a considerable fraction of total carbon
assimilation, occasionally even exceeding it (Tab. 1). In soils, the dark DIC fixation rates
which were attributed mainly to the activity of heterotrophs amounted to 0.04-39% of the
overall respiration rate (Tab. 1). Dark DIC fixation rates range from 36 ng C to 23.6 µg C $g^{-1}$ $d^{-1}$
$^{1}$ ranging over three orders of magnitude (Tab. 1). The contribution of heterotrophically
fixed DIC to biomass carbon of microbes ranged from 0.2-1.1% in temperate forest soil
(Akinyede et al. 2020), 0.2-4.6% in temperate forest and field soils (Santruckova et al. 2005),
to 7% in arable soil (Miltner et al. 2004). Santruckova et al. (2005) estimated the overall
heterotrophic $CO_2$ fixation to be even higher, i.e., 1.9-11.3% taking into account that the
labile fraction of the biodegradable organic carbon resulted from metabolites released by
spilling reactions of microorganisms due to a limitation in inorganic nutrients or due to the
presence of highly reduced energy-rich carbon sources (e.g. Tempest et al. 1992). A

contribution of heterotrophic $CO_2$ fixation to biomass carbon of 6.5±2.8% was found in drinking water biofilms and activated sludge (Roslev et al. 2004).

**Tab. 1.:** Dissolved inorganic carbon (DIC) assimilation rates from a range of aquatic (marine and limnic) and soil environments. Dark carbon fixation (DCF) is shown as fraction of either bacterial heterotrophic production (BP) or respiration. Original data were converted to similar units whenever possible to allow comparison.

| Aquatic ecosystems | Depth [m] | DIC fixation [µg C $L^{-1}$ $d^{-1}$] | BP [µg $C^{-1}$ $d^{-1}$] | DCF/BP [%] | Source | Remarks |
|---|---|---|---|---|---|---|
| Arctic | Seawater cultures | 0.5-2.3 | 0.4-2.5 | 100% | Alonso-Saéz et al. 2010 | Only potential for DCF |
| Mediterranean Sea | 4900 | 0.096 ± 0.02 | 0.048 | 200% | Yakimov et al. 2014 | Only anaplerotic |
| Tropical South China Sea | 200-1500 | 0.72-1.68 | 0.48- 4.8 | 40-105% | Zhou et al. 2017 | Probably a large fraction anaplerotic |
| Tropical Estuary | 1-18 | 4.8-14.4 | 55.2-1142 | 1.3-9% | Signori et al. 2018 | Probably mostly anaplerotic |
| Eutrophic lagoon | 1-5 | 206 | | | Lliros et al. 2011 | Probably mostly anaplerotic |
| Boreal lakes sediments | 1-3 | 13.2-48 mg C $m^{-2}$ $d^{-1}$ | BP 96-216 mg C $m^{-2}$ $d^{-1}$ | 8.4-37.4% | Santoro et al. 2013 | Probably a large fraction anaplerotic |
| Tropical lakes sediments | 1-3 | 0.12-20.4 mg C $m^{-2}$ $d^{-1}$ | BP 14.4- 583 mg C $m^{-2}$ $d^{-1}$ | 0.4-80.4% | Santoro et al. 2013 | Probably a large fraction anaplerotic |
| Deep granitic groundwater biofilms | 812-1240 | 0. 2-2 µg C $m^{-2}$ $d^{-1}$ | n.d. | n.d. | Ekendahl and Pedersen 1994 | Probably a large fraction anaplerotic |
| **Terrestrial ecosystems** | | DIC fixation [µg C $g^{-1}$ $d^{-1}$] | R [µg $CO_2$-C $g^{-1}$ $d^{-1}$] | DCF/R [%] | | |
| Temperate forest soil | 0-0.7 | 0.036-0.32 | 0.95-19.1 | 1.2-3.9% | Spohn et al. 2019 | [13]C label mainly in AA, indicating anaplerosis |
| | 0-1 | 0.06-0.86 | n.d. | n.d. | Akinyede et al. 2020 | Dominance of heterotrophs |
| Temperate agricultural soil | 0-0.3 | 0.26 | .63 | 2.7% | Miltner et al. 2004 | Probably a large fraction anaplerotic |
| | 0-0.3 | 0.19 | 9.82 | 1-5% | Miltner et al. 2005 | DCF mainly driven by aerobic heterotrophs |
| Range of temperate forest & field soils | 0.05-0.15 | 1.82-23.6* | 0.65-9.16 | 3-39% | Šantrůčková et al. 2005 | Probably a large fraction anaplerotic |
| | 0-0.15 | 0.035-0.4 | n.d. | n.d. | Nel and Cramer 2019 | Probably mostly anaplerotic |
| Arctic tundra soils | | 0.04-0.08 | 0.79-10.7 | 0.04-16% | Šantrůčková et al. 2018 | Anaplerotic enzymes comprised the majority of carboxylase genes. |

*Values taken from Table 2 in Akinyede et al. 2020
n.d. not determined

*Carbon biomass stock originating from heterotrophic $CO_2$ fixation*

While it is difficult to derive global estimations from the few studies that measured heterotrophic $CO_2$ fixation rates in marine, limnic and terrestrial ecosystems, we may use a conservative approach assuming that at least 1-5% of carbon biomass of all heterotrophs originates from anaplerotic DIC fixation. Earth's total living biomass is estimated to amount to about 499 – 738 Pg C, of which approx. 451 – 653 Pg C is photoautotrophic biomass (Bar-On et al. 2018). Heterotrophic biomass thus contributes 47 – 85 Pg C (Table SI-1). The, uncertainties of the estimates of heterotrophic biomass of the terrestrial subsurface, however, are high (Whitman et al. 1998, McMahon and Parnell 2014, Bar-On et al. 2018). Nevertheless, following this line of evidence anaplerotic $CO_2$ fixation contributes between 0.5 – 5 Pg C to the living biomass.

*Carbon flux related to heterotrophic $CO_2$ fixation*

In terms of annual global heterotrophic production rates, oceans and the terrestrial subsurface (including soils) are the main habitats of heterotrophic $CO_2$ fixation (Cole et al. 2002; Magnabosco et al. 2018) (Table SI-2). Recently, Akinyede et al. (2020) estimated a global dark $CO_2$ fixation rate of all temperate forest soils of $0.26 \pm 0.07$ Pg C $yr^{-1}$. We calculated a global heterotrophic C production of 34 – 245 Pg C $yr^{-1}$, which would translate into 0.34 – 12.3 Pg of DIC bound by heterotrophic $CO_2$ fixation each year. Interestingly, these numbers are consistent with the recently calculated contribution of $CO_2$ fixation for the integrated epipelagic ocean of ca. 1.2– 11 Pg C $yr^{-1}$ (Baltar and Herndl 2019). This is a significant carbon flux amounting to 0.3-14% of the global net amount of carbon produced annually by photoautotrophs (90 – 110 Pg C $yr^{-1}$; Ciais et al. 2013).

Our estimates are subject to a high uncertainty, which, on the one hand, results from the dependency of the extent of heterotrophic $CO_2$ fixation on the organic carbon oxidized and, on the other hand, on the predominant environmental conditions. Moreover, data on terrestrial and marine subsurface environments, although large in dimension, are scarce. For these environments, no detailed information on the abundance, growth (yield) and metabolic activity of microbial communities is available, particularly with increasing depth. Most of the deeper subsurface environments, even when harboring considerable living biomass, do not participate in the global carbon cycle on a short and medium time scales (years to decades), but rather in centennial to geological time scales. Nevertheless, in order to provide a first estimate and to be able to roughly evaluate the relevance of heterotrophic $CO_2$ fixation for all habitats of high uncertainty (e.g. the continental subsurface) we adopted a conservative approach (see also Tables SI-1 and SI-2).

## 5. Conclusions

Current models of carbon cycling and carbon sequestration do not account for heterotrophic $CO_2$ fixation (Gruber et al. 2004, Le Quéré et al. 2009). Despite the uncertainties in the data on heterotrophic biomass and production rates for some habitats (e.g. the terrestrial subsurface), the numbers presented here represent the first attempt to quantify the global contribution and relevance of heterotrophic $CO_2$ fixation to carbon cycling. Our results indicate that heterotrophs significantly contribute to global $CO_2$ fixation – especially (although not restricted to) in habitats experiencing elevated $CO_2$ concentrations and/or lacking a sufficient supply of degradable organic carbon. In specific environments, this may explain the mismatch between autotrophic C input, consumption, and sequestration that has been observed in marine systems (Baltar et al. 2009, Burd et al. 2010, Reinthaler et al. 2010, Morán et al. 2007, Hoppe et al. 2002, Tait and Schiel 2013). Particularly in aphotic habitats (which outnumber the photic habitats in both size and volume) such as the dark ocean, subseafloor sediments, soils, as well as the sediments and rocks of the terrestrial subsurface (Miltner et al. 2004, Miltner et al. 2005, Yakimov et al. 2014, Wegener et al. 2012), carbon cycling needs to be re-evaluated taking into account anaplerotic $CO_2$ fixation and other inorganic carbon uptake pathways in heterotrophs. In subseafloor sediments, wetlands and marshes, as well as in other habitats where methane oxidation is a key process, a large fraction (10-50%) of heterotrophic biomass potentially originates from heterotrophic DIC fixation. Recently, a time-series study showed a tendency towards higher ratios of dark to light DIC fixation in the top half of the euphotic layer (0– 65 m) in the years 2012-2019 than in the preceding years (data started in 1989), which was linked to oceanographic changes (i.e., a deepening of the mixed zone) (Baltar et al., 2019). Moreover, the metabolic theory of ecology posits that heterotrophic metabolism increases more than gross primary production in the ocean in response to warming (see Baltar et al., 2019 and reference therein), which might also make heterotrophic DIC fixation relatively more important in a warmer ocean. In the light of global warming leading to an extensive thawing of permafrost soils and providing new habitats for methanotrophs, these processes are expected to become more important in the future. Hence, the potential contribution of heterotrophic $CO_2$ fixation under climate change conditions clearly deserves further investigations.

## Author contributions

A.B., M.E. and C.G. conceived the idea for the manuscript. A.B., G.J.H. and C.G. wrote the manuscript. M.S.F., M.E., M.A. F.B. and T.R. substantially commented on and edited the manuscript. M.A., M.S.F. and C.G. did the literature search on available global carbon data. C.G. and M.A. performed the estimation of heterotrophic $CO_2$ fixation on a global scale.


**Acknowledgments**
We acknowledge B.B. Jørgensen for commenting on an earlier draft of the manuscript. We
thank R. Thauer and W. Eisenreich for fruitful discussions on heterotrophic $CO_2$ fixation.
Financial support was provided by the Wittgenstein Prize (Austrian Science Fund, project
number Z194-B17), by the European Research Council under the European Community's
Seventh Framework Program (FP7/2007-2013) / ERC grant agreement No. 268595 (MEDEA
project) and the Austrian Science Fund (P 28781-B21) to G.J.H. Financial support was further
provided by the Helmholtz Center Munich to A.B., M.E., M.S.F. and C.G.

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
