# Peer review of "significant but invisible flux in environmental carbon cycling"

_Biogeosciences, 2020_

## Referee Comment (RC1) · Anonymous Referee #1 · 3 Jan 2021

I was very excited to see this synthesis and review paper on heterotrophic CO2 fixation because heterotrophic CO2 fixation is currently not well understood despite indications that it is quantitatively important in several ecosystems. I started reading the manuscript with great interest, but unfortunately, found the manuscript increasingly disappointing for the following reasons.

(1) Simplistic estimation of global fluxes: The main synthesis work done by the authors in the present study is summarized in the two tables of the manuscript. Table 1 gives the global standing stock of organic carbon in living biomass and the contribution from anaplerotic CO2 fixation. In this table, the authors compiled data on C stocks in different biomass pools. Next, they multiplied the stocks of biomass of heterotrophs by 0.02 and 0.08 and the stocks of biomass of photoautotrophs by 0.01 and 0.05 in order to estimate the contribution from anaplerotic CO2 fixation. Likewise, in Table 2 the authors compiled data on the annual global heterotrophic carbon biomass production for different ecosystems and multiply it by 0.02 and 0.08 in order to estimate the contribution from anaplerotic CO2 fixation. In the first part of the manuscript, the authors point out that there is large uncertainty concerning the rates of microbial heterotrophic CO2 fixation and the underlying metabolic pathways. I agree with this view, and was very surprised to see that the authors estimate global rates of heterotrophic CO2 fixation based on a simple multiplication. I find it highly questionable to base a review paper on this kind of back on the envelop calculation, and I do not see the value of this estimation given how little we know about the size of the flux in different ecosystems.

(2) Lack of synthesis of empirical data: From a review and synthesis paper on heterotrophic CO2 fixation, I would expect a review of empirically determined CO2 fixation rates. Unfortunately, this is lacking in the present manuscript. For some inspiration, the authors should have a look at the recent study by Akinyede et al., 2020 who nicely compiled data on heterotrophic CO2 fixation in soils in a table that gives a good overview.

(3) Unclear scope and unbalanced review of literature: According to the title and the Introduction of the manuscript, the topic of the manuscript is heterotrophic fixation of inorganic carbon. However, the manuscript focuses very strongly on CO2 fixation through anaplerotic reactions and pays less attention to other pathways of heterotrophic CO2 fixation. More importantly, the manuscript concentrates strongly on literature about aquatic ecosystems and largely ignores literature about heterotrophic CO2 fixation in terrestrial ecosystems. This is problematic given that the authors state that the manuscript has a global scope.

(4) Unfocused and unclear figures: I like Fig. 2 because it gives a good overview over different pathways. However, I did not understand the purpose of Fig. 1 and the reason why it only shows one pathway. I guess that the purpose of the figure is mainly to show

that the CO2 can be derived either from cell-internal or cell-external processes. While this might be important for the estimation of fluxes, I'm not fully convinced that this requires a separate figure. More importantly, I did not understand Fig. 3. According to the caption the figure shows how much CO2 is fixed when microbes feed on different carbon substrates. However, somehow the gist seems to be that this is always 8% as indicated by the grey area. What I did not understand is the meaning of the black line and why it goes up all the way to 45%. Maybe a legend would be helpful here?

I am sorry that I cannot provide a more positive review of this synthesis study. I hope that my criticism is clear and that it provides some guidance on how to improve the synthesis study.

References

Akinyede, R., Taubert, M., Schrumpf, M., Trumbore, S., & Küsel, K. (2020). Rates of dark CO2 fixation are driven by microbial biomass in a temperate forest soil. Soil Biology and Biochemistry, 150, 107950.

---

## Referee Comment (RC2) · Anonymous Referee #2 · 12 Feb 2021

Broadly speaking, this is a timely review on carbon dioxide fixation by heterotrophs, a process that is likely significant in many biomes, yet typically overlooked in biogeochemical studies. Even if I appreciate that the authors bring up this topic, the manuscript suffers from being too broad in scope and my first and most important recommendation would be to limit the synthesis to the marine environment as this seems to be where most of the relevant cited papers are from. This would make the manuscript more coherent and informative while also limiting the speculative elements. Specific comments provided below.

Figure 1 is overall appealing and clear, but what is not evident is where the data supporting the quantitative information comes from. It is stated that the thickness of the arrows represents the relative contribution to fluxes. This is likely context depending and might be relevant in some (but certainly not all) biological systems and it needs to be clearly specified (and referenced) how this quantitative information was obtained and under what circumstances they are valid. This will most likely also vary between different heterotrophs and this need to come across in the figure.

I have similar concerns for figure 3, as it is not evident whether the (again) quantitative information derive from purely theoretical reasoning or if there is empirical evidence? In the latter case I of course want to know under what conditions these results are valid and how they were obtained. References needed!

In line with these comments, I found that a lot of the inferences and assumptions made for natural communities are based on early work with pure cultures. It is not evident that such extrapolations are valid. A more critical discussion about this is needed.

One fundament of the arguments made is the strong link between the degree of reduction of the substrate being metabolized in relation to the same metric for (average) biomass. I think this is an interesting and potentially very useful approach, but biomass is not only carbon. Also other elements (nitrogen, sulfur, etc) are assimilated in large quantities and these can also have different degree of reduction (ammonia and nitrate as an example). This can of course also have a major impact on the need for anaplerotic $CO_2$ fixation but is completely overlooked in this synthesis.

I strongly object to the simplistic way of estimating the quantitative significance of heterotrophic $CO_2$ fixation as presented in tables 1-2. This is way too speculative. This type of estimates should not be extrapolated beyond the environments where the process has actually been quantified with reliable modern methods. Assuming an equal percentage of heterotroph biomass to come from inorganic carbon fixation will most certainly be misleading, especially moving from the reasonably well studied oceanic waters to terrestrial and deep biosphere biomes where very different conditions may

prevail. I would strongly encourage the authors to show some restraint here. One solution could (again) be to change the stated scope of the review to focus on marine waters where it might be at least somewhat reasonable to make such assumptions.

The cited papers include both modern and old cited papers (some from the 1960's) and I doubt methods and approaches for quantifying and documenting heterotrophic carbon fixation have remained the same. Methodological constraints and biases may play a major role here as the typically rather slow anaplerotic carbon fixation and other heterotrophic bacterial $CO_2$ fixation can be easily masked (or "contaminated") by other metabolic processes. Some critical review and account of the different methodological approaches used in the cited work would have been very valuable and useful for the reader. Are results from the 1960's more or less trustworthy that results obtained 50-60 years later?

I find the arguments about carbon use efficiency confusing (line 133-150). While the issues with using oxygen removal to deduce respiratory changes in carbon dioxide is quite well known and recognized in literature, but this may surely lead to both over-estimates and underestimates of $CO_2$ production depending on the particular context. With that in mind it is not clear to me how this leads the authors to the following conclusion: "Collectively, with respect to C cycling, heterotrophic $CO_2$ fixation and the carbon flux from the inorganic pool into heterotrophic biomass can be regarded as a process more important than hitherto assumed". This may not be true for all marine conditions and certainly not for systems such as the deep biosphere, soils, freshwaters and other systems where other constrains may prevail and where the methodological tradition differ.

Some additional specific comments:

Line 42: Actually, these rates have been quantified in numerous studies in various ecosystems and for various organisms. Please revise.

Line 94-95: An alternative for adjusting the degree of reduction of the substrate to that

of average biomass is of course to oxidize the organic substrates in respiratory processes and at the same time gain energy for cellular processes and assembly. This is quite evident and the authors also make this clear elsewhere in the text. Nevertheless, the current statement made here needs to be adjusted.

Line 153-155: Confusing statement as chemolitoautotrophs which dominate in many dark environments are also autotrophs. Again: focus on marine systems and by making some more specific statements for this biome, the story will be more substantiated and to the point

Line 152-207: There is a quite substantial body on literature about this, but here the discussion is more or less exclusively about work in the oceans. See earlier comments about this bias in referenced literature.

Line 215-218: To make this assumption beyond oceans you would also need to present data demonstrating that the 2-8% is also valid for the other biomes.

Line 234: "scares" should be "scarce"

Line 340: is "exemplarily" really the right word here?

Line 353: Why introduce the abbreviation? Is it used anywhere else?

Line 353-360: This is very speculative and relies on the assumption that heterotrophic $CO_2$ fixation is and will remain a constant proportion of heterotrophic production. Where is the evidence for that? I do not find this likely given the current information we have about the metabolic diversity of microbial communities. The argument about methanotrophs is more credible as they evidently use much more inorganic carbon in their anabolic processes.

---

## Author Comment (AC1) · 26 Apr 2021

We thank Rev 1 for the valuable critisism and recommendations.Anonymous Referee #1

Anonymous Referee #1 I was very excited to see this synthesis and review paper on heterotrophic CO2 fixation because heterotrophic CO2 fixation is currently not well understood despite indications that it is quantitatively important in several ecosystems. I started reading the manuscript with great interest, but unfortunately, found the manuscript increasingly disappointing for the following reasons. (1) Simplistic estimation of global fluxes: The main synthesis work done by the authors in the present study is summarized in the two tables of the manuscript. Table 1 gives the global standing stock of organic carbon in living biomass and the contribution from anaplerotic $CO_2$ fixation. In this table, the authors compiled data on C stocks in different biomass pools. Next, they multiplied the stocks of biomass of heterotrophs by 0.02 and 0.08 and the stocks of biomass of photoautotrophs by 0.01 and 0.05 in order to estimate the contribution from anaplerotic $CO_2$ fixation. Likewise, in Table 2 the authors compiled data on the annual global heterotrophic carbon biomass production for different ecosystems and multiply it by 0.02 and 0.08 in order to estimate the contribution from anaplerotic $CO_2$ fixation. In the first part of the manuscript, the authors point out that there is large uncertainty concerning the rates of microbial heterotrophic $CO_2$ fixation and the underlying metabolic pathways. I agree with this view, and was very surprised to see that the authors estimate global rates of heterotrophic $CO_2$ fixation based on a simple multiplication. I find it highly questionable to base a review paper on this kind of back on the envelop calculation, and I do not see the value of this estimation given how little we know about the size of the flux in different ecosystems.

Reply: We are sorry for the disappointment. After careful reflection and thinking we revised the MS according this point of criticism in the following way. The major focus of the MS is not anymore on the global estimates. We included now a substantial amount of new information including quantitative data from different environments (aquatic systems and soils). See also the new table 1. In the last section of the MS we still present a first global estimation, however, now it is even more conservative (we assume 1-5% of biomass carbon to originate from analplerotic DIC fixation), and the respective table have been moved into the Supplimentary Information. The big uncertainties are repeatedly mentioned in the MS.

(2) Lack of synthesis of empirical data: From a review and synthesis paper on heterotrophic $CO_2$ fixation, I would expect a review of empirically determined $CO_2$ fixation rates. Unfortunately, this is lacking in the present manuscript. For some inspiration, the

authors should have a look at the recent study by Akinyede et al., 2020 who nicely compiled data on heterotrophic CO2 fixation in soils in a table that gives a good overview.

Reply: Thank you for this valid suggestion. After consultation of the table in Akinyede et al. (2020), which reports 'dark' CO2 fixation rates, we have compiled a synthesis table including all empirical data we could find. Our table compiles only studies where a significant contribution of DIC fixation from heterotrophs is reported.

(3) Unclear scope and unbalanced review of literature: According to the title and the Introduction of the manuscript, the topic of the manuscript is heterotrophic fixation of inorganic carbon. However, the manuscript focuses very strongly on CO2 fixation through anaplerotic reactions and pays less attention to other pathways of heterotrophic CO2 fixation. More importantly, the manuscript concentrates strongly on literature about aquatic ecosystems and largely ignores literature about heterotrophic CO2 fixation in terrestrial ecosystems. This is problematic given that the authors state that the manuscript has a global scope.

Reply: We agree with the reviewer that the MS's content was unbalanced. The strong focus on marine habitats has now been balanced by new information from limnic and terrestrial (soils) ecosystems incorporated, not only in the new Table 1 but also in the text sections. The second point of criticism , i.e. exclusive focus on anaplerosis, is difficult to address. While we have included more information and citations with respect to carboxylases others than involved in anaplerosis, there is no quantitative data on CO2 fixation rates by these other enzymes and pathways. That's why our conservative estimations base exclusively on anaplerosis which is ubiquitously present in all organisms.

(4) Unfocused and unclear figures: I like Fig. 2 because it gives a good overview over different pathways. However, I did not understand the purpose of Fig. 1 and the reason why it only shows one pathway. I guess that the purpose of the figure is mainly to show that the CO2 can be derived either from cell-internal or cell-external processes. While

this might be important for the estimation of fluxes, I'm not fully convinced that this requires a separate figure. More importantly, I did not understand Fig. 3. According to the caption the figure shows how much CO2 is fixed when microbes feed on different carbon substrates. However, somehow the gist seems to be that this is always 8% as indicated by the grey area. What I did not understand is the meaning of the black line and why it goes up all the way to 45%. Maybe a legend would be helpful here? I am sorry that I cannot provide a more positive review of this synthesis study. I hope that my criticism is clear and that it provides some guidance on how to improve the synthesis study.

Reply: Thank you for this comment. Yes, Fig. 1 was intended to highlight the possibility of a simultaneous fixation of internal and external DIC sources. However, the reviewer is absolutely right, it is too simplistic in ignoring other pathways. We deleted Fig. 1. In Figure 3 (now Fig. 2) we revised the caption text. Now what's seen in the figure should be clear. In detail, der red arrows depict empirical measurements on how much biomass carbon was contributed from DIC fixation. The grey area highlights the general range (1-8%) found in various studies. Right from the dashed line, further carboxylase reactions contribute, beyond the 1-8% range, to the much higher assimilation of DIC. See caption in new Fig. 2.

References Akinyede, R., Taubert, M., Schrumpf, M., Trumbore, S., & Küsel, K. (2020). Rates of dark CO2 fixation are driven by microbial biomass in a temperate forest soil. Soil Biology and Biochemistry, 150, 107950.

---

## Author Comment (AC2) · 26 Apr 2021

We thank Rev 2 for the valuable critisism and recommendations.

Anonymous Referee #2 Broadly speaking, this is a timely review on carbon dioxide fixation by heterotrophs, a process that is likely significant in many biomes, yet typically overlooked in biogeochemical studies. Even if I appreciate that the authors bring up this topic, the manuscript suffers from being too broad in scope and my first and most important recommendation would be to limit the synthesis to the marine environment as this seems to be where most of the relevant cited papers are from. This would

make the manuscript more coherent and informative while also limiting the speculative elements. Specific comments provided below.

Reply: We appreciate this overall comment. It is agreed that based on the original content and structure of our MS, one may derive this conclusion. However, the attempt was to not focus on marine systems. Instead of limiting the synthesis to the marine environment, we took a big effort to balance the information provided from marine studies with information from limnic and terrestrial (soils) ecosystems. The synthesis paper now combines all quantitative data on heterotrophic DIC fixation from aquatic and terrestrial habitats we could find. See new Table 1.

Figure 1 is overall appealing and clear, but what is not evident is where the data supporting the quantitative information comes from. It is stated that the thickness of the arrows represents the relative contribution to fluxes. This is likely context depending and might be relevant in some (but certainly not all) biological systems and it needs to be clearly specified (and referenced) how this quantitative information was obtained and under what circumstances they are valid. This will most likely also vary between different heterotrophs and this need to come across in the figure.

Reply: Thank you for this comment. We realized that Fig. 1 while being appealing is too simplistic and rather represents a certain situation/organisms than a general pattern. Because of similar comments received from Rev 1, we decided to remove Fig 1. To our opinion, it is not essential and all information is somehow provided also in the text.

I have similar concerns for figure 3, as it is not evident whether the (again) quantitative information derive from purely theoretical reasoning or if there is empirical evidence? In the latter case I of course want to know under what conditions these results are valid and how they were obtained. References needed! In line with these comments, I found that a lot of the inferences and assumptions made for natural communities are based on early work with pure cultures. It is not evident that such extrapolations are valid. A more critical discussion about this is needed. One fundament of the

arguments made is the strong link between the degree of reduction of the substrate being metabolized in relation to the same metric for (average) biomass. I think this is an interesting and potentially very useful approach, but biomass is not only carbon. Also other elements (nitrogen, sulfur, etc) are assimilated in large quantities and these can also have different degree of reduction (ammonia and nitrate as an example). This can of course also have a major impact on the need for anaplerotic $CO_2$ fixation but is completely overlooked in this synthesis.

Reply: The comments are appreciated. First of all, we revised the caption of the former Fig. 3, which is Figure 2 now. A much better and clearer explanation is provided. The figure contains real data (e.g. the red arrows depicting at the fraction of biomass carbon contributed from DIC fixation in different microbes when growing on different organic carbon sources) and in the new text added, the aspect of additional DIC fixation along with the oxidation of highly reduced organic carbon sources was elaborated more extensively. Several new references have been added. It is true that also other elements with species of differing degree of reduction (e.g. ammonia and nitrate) are assimilated. However, the overall amount is general one order of magnitude less. We do not think that this will interfere with the DIC fixation patterns by heterotrophs introduced here. In case of chemoautotrophy (e.g. DIC fixation in combination of ammonia oxidation by nitrifyers) of course, the amount of DIC fixed does play a role in biomass production. This is already mentioned at several spots in the MS.

I strongly object to the simplistic way of estimating the quantitative significance of heterotrophic $CO_2$ fixation as presented in tables 1-2. This is way too speculative. This type of estimates should not be extrapolated beyond the environments where the process has actually been quantified with reliable modern methods. Assuming an equal percentage of heterotroph biomass to come from inorganic carbon fixation will most certainly be misleading, especially moving from the reasonably well studied oceanic waters to terrestrial and deep biosphere biomes where very different conditions may prevail. I would strongly encourage the authors to show some restraint here. One solution could (again) be to change the stated scope of the review to focus on marine waters where it might be at least somewhat reasonable to make such assumptions.

A similar comment was received by Rev 1. Thank you for addressing this critical issue. As already mentioned above, we have changed the following. First, we recalculated our global estimates with 1-5% of heterotrophically fixed DIC contributing to biomass carbon. This leads to a more conservative estimation. All studies conducted, as well as theoretical calculations, underline that a minimum of 1% can be expected for all kind of organisms independent of the habitat they live in, and 5% is still very reasonable and can be understood of an upper value. However, to move the global estimates and the two tables out of the focus, we provide them now as Supplementary Information, and the discussion on global carbon stock and assimilation rates from anaplerosis is only a brief text section at the end of the MS. Moreover we included a pile of data on quantitative estimates from individual habitats intending a balance between information from aquatic and terrestrial systems; see new Table 1.

Reply: The cited papers include both modern and old cited papers (some from the 1960's) and I doubt methods and approaches for quantifying and documenting heterotrophic carbon fixation have remained the same. Methodological constraints and biases may play a major role here as the typically rather slow anaplerotic carbon fixation and other heterotrophic bacterial $CO_2$ fixation can be easily masked (or "contaminated") by other metabolic processes. Some critical review and account of the different methodological approaches used in the cited work would have been very valuable and useful for the reader. Are results from the 1960's more or less trustworthy that results obtained 50-60 years later? Thank you for this comment. We actually have thought of this aspect when putting the information together. Our impression is that most of the early studies, done by biochemists, microbiologists and physiologists, deliver reliable information, with the weakness that data come from only laboratory experiments with microbial strains and tissue cultures. With some of the modern ecological studies, it is less clear what have been measured, as dark $CO_2$ fixation includes both chemoautotrophs and heterotrophs as actors. Honestly, we think that this issue is not really of relevance, because the numbers on heterotrophic $CO_2$ fixation as contribution to biomass carbon are always in the same range (about 1-8%, with some higher values too), independent from the time of the experiments or studies. In other words, we did not detect a systematic change in values from older studies to modern ones. We conclude that a separate discussion about this aspect is thus not necessary.

I find the arguments about carbon use efficiency confusing (line 133-150). While the issues with using oxygen removal to deduce respiratory changes in carbon dioxide is quite well known and recognized in literature, but this may surely lead to both over-estimates and underestimates of $CO_2$ production depending on the particular context. With that in mind it is not clear to me how this leads the authors to the following conclusion: "Collectively, with respect to C cycling, heterotrophic $CO_2$ fixation and the carbon flux from the inorganic pool into heterotrophic biomass can be regarded as a process more important than hitherto assumed". This may not be true for all marine conditions and certainly not for systems such as the deep biosphere, soils, freshwaters and other systems where other constrains may prevail and where the methodological tradition differ.

Reply: Sorry that our line of argumentation was not clear. What we wanted to say is that frequently the amount of $CO_2$ produced by oxidation when being calculated via the amount of $O_2$ consumed, does not take the 'recycling' of $CO_2$ via anaplerosis into account. Moreover, the contribution of anaplerosis and other carboxylation reaction not linearly correlate with $O_2$ consumption. I hope we have made this now clear in the revised text. As done in Akinyede et al. 2020, one could calculate the $CO_2$ not emitted from a habitat due to heterotrophic assimilation. Strict measurements of $O_2$ comsumption (common in aquatic environments) and respiration (common in soil environments) do not provide this information.

Some additional specific comments: Line 42: Actually, these rates have been quantified in numerous studies in various ecosystems and for various organisms. Please

revise.

Reply: Yes, true. This is exactly what we show with our MS. We changed this sentence accordingly.

Line 94-95: An alternative for adjusting the degree of reduction of the substrate to that of average biomass is of course to oxidize the organic substrates in respiratory processes and at the same time gain energy for cellular processes and assembly. This is quite evident and the authors also make this clear elsewhere in the text. Nevertheless, the current statement made here needs to be adjusted.

Reply: We are sorry but we did not fully understand this comment. What we say is: If the organic matter that is oxidized in respiratory processes (aerobic and anaerobic) is more reduced that the organisms' biomass carbon further $CO_2$ is fixed in addition to what is fixed already in anaplerotic reactions. As an extreme example methanotrophy is mentioned. Energy gained by the catabolic reaction (respiration process) is used to build biomass. A special case is spilling reactions where organic carbon is in excess but growth is limited by essential nutrients. We have mentioned that in the MS. We hope, we are clear now.

Line 153-155: Confusing statement as chemolitoautotrophs which dominate in many dark environments are also autotrophs. Again: focus on marine systems and by making some more specific statements for this biome, the story will be more substantiated and to the point.

Reply: The sentence war corrected to avoid confusion.

Line 152-207: There is a quite substantial body on literature about this, but here the discussion is more or less exclusively about work in the oceans. See earlier comments about this bias in referenced literature.

Reply: We agree. We have now implemented new information on limnic systems and terrestrial environments to balance the synthesis. A focus on marine systems was

never intended.

Line 215-218: To make this assumption beyond oceans you would also need to present data demonstrating that the 2-8% is also valid for the other biomes.

Reply: This is a valid comment. We have now incorporated data from all different kinds of organisms and different biomes to better support our assumptions.

Line 234: "scares" should be "scarce"

Reply: Changed accordingly.

Line 340: is "exemplarily" really the right word here?

Reply: Changed accordingly.

Line 353: Why introduce the abbreviation? Is it used anywhere else?

Reply: Abbreviation deleted.

Line 353-360: This is very speculative and relies on the assumption that heterotrophic $CO_2$ fixation is and will remain a constant proportion of heterotrophic production. Where is the evidence for that? I do not find this likely given the current information we have about the metabolic diversity of microbial communities. The argument about methanotrophs is more credible as they evidently use much more inorganic carbon in their anabolic processes.

Reply: We think, a minimum of DIC is fixed by every heterotrophic organism. This allows a first estimation. We agree that at certain physiological and environmental conditions it will be more, but never less that 1% of biomass carbon which is our minimum assumption. Evidence comes from numerous experiments with microbial strains, tissue cultures, metazoan, and also from a number of field studies, including soils, drinking water biofilms, activated sludge, to give some examples. We accordingly revised the text to include this information.

---

## Author Comment (AC3) · 26 Apr 2021

**Reviews and syntheses: Heterotrophic fixation of inorganic carbon – significant but invisible flux in environmental carbon cycling**

**Supplementary Information**

Alexander Braun[1], Marina Spona-Friedl[1], Maria Avramov[1], Martin Elsner[1,2], Federico Baltar[3], Thomas Reinthaler[3], Gerhard J. Herndl[3,4] & Christian Griebler[1,3]*

[1] Helmholtz Zentrum München, Institute of Groundwater Ecology, Ingolstaedter Landstrasse 1, D-85764 Neuherberg, Germany

[2] Technical University of Munich, Department of Analytical Chemistry and Water Chemistry, Munich, Germany

[3] University of Vienna, Department of Functional and Evolutionary Ecology, Althanstrasse 14, 1090 Vienna, Austria

[4] Department of Marine Microbiology and Biogeochemistry, Royal Netherlands Institute for Sea Research, Utrecht University, PO Box 59, 1790 AB Den Burg, The Netherlands

* Author for correspondence: christian.griebler@univie.ac.at

**Table SI-1:** Global standing stock of organic carbon in living biomass and contribution from anaplerotic $CO_2$
fixation (only anaplerosis is considered here; other mechanisms of heterotrophic $CO_2$ fixation were neglected).
In heterotrophs, a conservative estimate of 1-5% of the cell carbon is assumed to originate from inorganic
carbon fixation (see references in text).

| Continental habitats | Carbon biomass [Pg C] | Carbon in biomass derived from anaplerotic $CO_2$ fixation [Pg C] | References for carbon biomass |
|---|---|---|---|
| Terrestrial animals | 0.6 | 0.006 – 0.03 | (Bar-On et al. 2018) |
| Soil fungi | 12 | 0.12 – 0.6 | (Bar-On et al. 2018) |
| Terrestrial protists | 1.6 | 0.016 – 0.8 | (Bar-On et al. 2018) |
| Soil prokaryotes (upper 100 cm of soil) | 23.2 | 0.23 – 1.16 | (Xu, Thornton and Post 2013) |
| Continental subsurface prokaryotes | 2.4 – 12.6* | 0.024 – 0.63 | (Magnabosco et al. 2018) |
| Heterotrophic prokaryotes in freshwater and saline inland surface waters | 0.013** | 0.00013 – 0.00065 | (Whitman et al. 1998) |
| **Marine and oceanic habitats** | | | |
| Marine Animals | 2 | 0.02 – 0.1 | (Bar-On et al. 2018) |
| Marine protists | 2 | 0.02 – 0.1 | (Bar-On et al. 2018) |
| Marine fungi | 0.3 | 0.003 – 0.015 | (Bar-On et al. 2018) |
| Marine planktonic heterotrophic prokaryotes | 1.4 – 3.5*** | 0.014 – 0.175 | (Whitman et al. 1998) |
| Subseafloor sedimentary prokaryotes | 1.5 – 22 | 0.015 – 1.1 | (Kallmeyer et al. 2012, Schippers et al. 2005) |
| Prokaryotes of the oceanic crust | 0.5 – 5 | 0.005 – 0.25 | (Bar-On et al. 2018) |
| **Total heterotrophic carbon biomass** | **47 – 85** | **0.47 – 4.96** | |

* Cell abundances ($2 – 6 \times 10^{29}$ cells) from Magnabosco et al. (2018) were converted into cell carbon using the
carbon conversion factors 12 fg C cell$^{-1}$ and 21 fg C cell$^{-1}$ (Wilhartitz et al. 2009, Griebler et al. 2002) for the
minimum and maximum values of the range, respectively. In favor of a conservative estimate, quite low carbon
conversion factors were used (at the lower end of the carbon content values for freshwater prokaryotic cells
reported in literature).

** Cell abundance ($2.3 \times 10^{26}$ cells) from Whitman et al. (1998) were converted into cell carbon using a carbon
conversion factor of 57 fg C cell$^{-1}$, which is the arithmetic mean of the minimum and maximum of a range of
values (6 to 107 fg C cell$^{-1}$) reported for freshwater lakes and rivers of different trophic states in literature
(Pedrós-Alió and Brock 1982, Bjørnsen 1986, Simon 1987, Lever et al. 2015).

*** Cell abundances were converted into cell carbon using the carbon conversion factors 12 fg C cell$^{-1}$ and
30 fg C cell$^{-1}$ (Fukuda et al. 1998) for the minimum and maximum values, respectively.

[revised manuscript text omitted]

del Giorgio, P. A., and Duarte, C. M.: Respiration in the open ocean. Nature, 420, 379-384, 2002.

Falkowski, P. G., Barber, R. T., and Smetacek, V.: Biogeochemical controls and feedbacks on ocean
primary production. Science, 281, 200-206, 1998.

Fukuda, R., Ogawa, H., Nagata, T., and Koike, I.: Direct determination of carbon and nitrogen
contents of natural bacterial assemblages in marine environments. Appl. Environ. Microbiol., 64,
3352-3358, 1998.

Griebler, C., Mindl, B., Slezak, D., and Geiger-Kaiser, M.: Distribution patterns of attached and
suspended bacteria in pristine and contaminated shallow aquifers studied with an in situ sediment
exposure microcosm. Aquat. Microb.Ecol., 28, 117-129, 2002.

Griebler, C., Hahn, H. J., Stein, H., Kellermann, C., Fuchs, A., Steube, C., Berkhoff, S., and Brielmann,
H.: Development of a biological assessment scheme and criteria for groundwater ecosystems
(Entwicklung biologischer Bewertungsmethoden und -kriterien für Grundwasserökosysteme). Report
to the German Federal Environmental Agency (UBA); UFOPLAN grant no. 3708 23 200, ISSN: 1862-
4804, 153 pp., 2014.

Hashimoto, S., Carvalhais, N., Ito, A., Migliavacca, M., Nishina, K., and Reichstein, M.: Global
spatiotemporal distribution of soil respiration modeled using a global database. Biogeosci., 12, 4121-
4132, 2015.

Kallmeyer, J., Pockalny, R., Adhikari, R. R., Smith, D. C., and D'Hondt, S.: Global distribution of
microbial abundance and biomass in subseafloor sediment. PNAS, 109, 16213-16216, 2012.

Kazumi, J., and Capone D. G.: Heterotrophic microbial activity in shallow aquifer sediments of Long
Island, New York. Microb. Ecol., 28, 19-37, 1994.

Kieft, T. L., and Simmons K. A.: Allometry of animal-microbe interactions and global census of animal-
associated microbes. Proceedings of the Royal Society of London B: Biol. Sci., 282, 1-8, 2015.

Lever, M. A., Rogers, K. L., Lloyd, K. G., Overmann, J., Schink, B., Thauer, R. K., Hoehler, T. M. &
Jørgensen, B. B.: Life under extreme energy limitation: a synthesis of laboratory- and field-based
investigations. FEMS Microbiol. Rev., 39, 688-728, 2015.

Magnabosco, C., Lin, L. H., Dong, H., Bomberg, M., Ghiorse, W., Stan-Lotter, H., Pedersen, K., Kieft, T.
L., van Heerden, E., and Onstott, T. C. The biomass and biodiversity of the continental subsurface.
Nature Geoscience, 11, 707-717, 2018.

Manzoni, S., Taylor, P., Richter, A., Porporato, A., and Ågren, G. I.: Environmental and stoichiometric
controls on microbial carbon-use efficiency in soils. New Phytologist, 196, 79-91, 2012.

McMahon, S., and Parnell J. Weighing the deep continental biosphere. FEMS Microbiol. Ecol., 87,
113-120, 2014.

Pedrós-Alió, C., and Brock T. D.: Assessing biomass and production of bacteria in eutrophic lake
Mendota, Wisconsin. Appl. Environ. Microbiol., 44, 203-218, 1982.

Potter, C. S., and Klooster S. A.: Interannual variability in soil trace gas($CO_2$, $N_2O$, NO) fluxes and
analysis of controllers on regional to global scales. Global Biogeochem. Cycl., 12, 621-635, 1998.

Prentice, I. C., G. D. Farquhar, M. J. R. Fasham, M. L. Goulden, M. Heimann, V. J. Jaramillo, H. S.
Kheshgi, C. Le Quéré, R. J. Scholes, and D. W. R. Wallace.: The carbon cycle and atmospheric carbon
dioxide. In Climate Change 2001: The Scientific Basis. Contribution of Working Group I to the Third
Assessment Report of the Intergovernmental Panel on Climate Change eds. J. T. Houghton, Y. Ding,
D. J. Griggs, M. Noguer, P. J. van der Linden, X. Dai, K. Maskell & C. A. Johnson, 183-237. Cambridge,
United Kingdom and New York, NY, USA: Cambridge University Press, 2001.

Schippers, A., Neretin, L. N., Kallmeyer, J., Ferdelman, T. G., Cragg, B. A., Parkes, R. J. & Jørgensen, B.
B.: Prokaryotic cells of the deep sub-seafloor biosphere identified as living bacteria. Nature, 433,
861-864, 2005.

Simon, M.: Biomass and production of small and large free-living and attached bacteria in Lake
Constance. Limnol. Oceanogr., 32, 591-607, 1987.

Thorn, P. M., and Ventullo, R. M. Measurement of bacterial growth rates in subsurface sediments
using the incorporation of tritiated thymidine into DNA. Microb. Ecol., 16, 3-16, 1988.

Whitman, W. B., Coleman, D. C. & Wiebe, W. J.: Prokaryotes: The unseen majority.PNAS, 95, 6578-
6583, 1998.

Wilhartitz, I. C., Kirschner, A. K. T., Stadler, H., Herndl, G. J., Dietzel, M., Latal, C., Mach, R. L., and
Farnleitner, A. H.: Heterotrophic prokaryotic production in ultra-oligotrophic alpine karst aquifers
and ecological implications. FEMS Microbiol. Ecol., 68, 287-299, 2009.

Xu, X., Thornton, P. E., and Post, W. M.: A global analysis of soil microbial biomass carbon, nitrogen
and phosphorus in terrestrial ecosystems. Global Ecol. .Biogeograph., 22, 737-749, 2013.

---

## Author Response (AR2)

Dear Editor, dear Reviewer 2

Thank you for your final comments and recommendations. All of the these final comments have been addressed and the submission is now ready for publication.

Rev2

(1) The meaning of the black line in figure 2 needs to be explained. This was brought up already in the first round of reviews, but I think the authors overlooked this specific comment.

*Reply: Explanation was added to the figure caption*

(2) Line 199-202: Reading this statement one get the impression that solar-exposed freshwaters and soils are insignificant biogeochemically, but this is of course not correct as volume does not necessarily scale with activity. The solar exposed surface waters and surface soils are disproportionately biologically active compared to most groundwaters and this should come across here.

*Reply: a sentence now underlines the disproportionality of biological activity of sunlit environments in comparison to the large-volume dark habitats.*

(3) Line 226-227: Awkward sentence. Quantitative data? Evidence? Strong support?

*Reply: We rephrased the sentence.*

Non-public comments from the Editor to the Author:

In addition to the small points raised by the reviewer, I do have one additional recommendation, which is to emphasize the difference - and the difficulties in differentiating - heterotrophic $CO_2$ from dark autotrophic $CO_2$ fixation. The data in the (new) Table 1 for example contain some comments in this direction, but I think it would help the reader if you point out also some methodological aspects and stress in the text to which extent the data in this Table refer to total dark $CO_2$ fixation (autotrophic + heterotrophic), or whether they specifically report heterotrophic $CO_2$ fixation.

*Reply: a short paragraph was added to critically emphasize the methodological aspects and uncertainties with heterotrophic carbon fixation data.*